# Adversarial representation learning for synthetic replacement of private attributes

## Abstract

Data privacy is an increasingly important aspect of many real-world big data analytics tasks. Data sources that contain sensitive information may have immense potential which could be unlocked using privacy enhancing transformations, but current methods often fail to produce convincing output. Furthermore, finding the right balance between privacy and utility is often a tricky trade-off. In this work, we propose a novel approach for data privatization, which involves two steps: in the first step, it removes the sensitive information, and in the second step, it replaces this information with an independent random sample. Our method builds on adversarial representation learning which ensures strong privacy by training the model to fool an increasingly strong adversary. While previous methods only aim at obfuscating the sensitive information, we find that adding new random information in its place strengthens the provided privacy and provides better utility at any given level of privacy. The result is an approach that can provide stronger privatization on image data, and yet be preserving both the domain and the utility of the inputs, entirely independent of the downstream task.

## 1 Introduction

Increasing capacity and performance of modern machine learning models lead to increasing amounts of data required for training them (Goodfellow et al., 2016). However, collecting and using large datasets which may contain sensitive information about individuals is often impeded by increasingly strong privacy laws protecting individual rights, and the infeasibility of obtaining individual consent. Giving privacy guarantees on a dataset may let us share data, while protecting the rights of individuals, and thus unlocking the large benefits for individuals and for society that big datasets can provide.

In this work, we propose a technique for selective obfuscation of image datasets. The aim is to provide the original data as detailed as possible while making it hard for an adversary to detect specific sensitive attributes. The proposed solution is agnostic to the downstream task, with the objective to make the data as private as possible given a distortion constraint. This issue has previously been addressed using adversarial representation learning with some success: a filter model is trained to obfuscate sensitive information while an adversary model is trained to recover the information (Edwards & Storkey, 2016). In the current work, we demonstrate that *it is easier to hide sensitive information if you replace it with something else*: a sample which is independent from the input data.

Aside from the adversary module, our proposed solution includes two main components: one filter model that is trained to remove the sensitive attribute, and one generator model that inserts a synthetically generated new value for the sensitive attribute. The generated sensitive attribute is entirely independent from the sensitive attribute in the original input image. Following a body of work in privacy-related adversarial learning we evaluate the proposed model on faces from the CelebA dataset (Liu et al., 2015), and consider, for example, the smile or gender of a person to be the sensitive attribute. The smile is an attribute that carries interesting aspects in the transformations of a human face. The obvious change reside close to the mouth when a person smiles, but also other subtle changes occur: eyelids tighten, dimples show and the skin wrinkles. The current work includes a thorough analysis of the dataset, including correlations of such features. These correlations make the task interesting and challenging, reflecting the real difficulty that may occur when anonymizing data. What is the right trade-off between preserving the utility as defined by allowing information about other attributes to remain, and removing the sensitive information?

In our setup, the adversary can make an arbitrary number of queries to the model. For each query another sample will be produced from the distribution of the sensitive data, while keeping as much as possible of the non-sensitive information about the requested data point.

## 2 RELATED WORK

Privacy-preserving machine learning has been studied from a number of different angles. Some work assumes access to a privacy-preserving mechanism, such as bounding boxes for faces, and studies how to hide people's identity by blurring (Oh et al., 2016a), removing (Orekondy et al., 2018) or generating the face of other people (Hukkelås et al., 2019) in their place. Other work assumes access to the utility-preserving mechanism and proposes to obfuscate everything except what they want to retain (Alharbi et al., 2019). This raises the question: how do we find the pixels in an image that need to be modified to preserve privacy with respect to some attribute?

Furthermore, Oh et al. (2016b) showed that blurring or removing the head of a person has a limited effect on privacy. The finding is crucial; *we cannot rely on modifications of an image such as blurring or overpainting to achieve privacy*. An adversarial set-up instead captures the signals that the adversary uses, and can attain a stronger privacy. Adversarial learning is the process of training a model to fool an adversary (Goodfellow et al., 2014). Both models are trained simultaneously, and become increasingly good at their respective task during training. This approach has been successfully used to learn image-to-image transformations (Isola et al., 2017; Choi et al., 2018), and synthesis of properties such as facial expressions (Song et al., 2017; Tang et al., 2019). Privacy-preserving adversarial representation learning utilize this paradigm to learn representations of data that hide sensitive information (Edwards & Storkey, 2016; Zhang et al., 2018; Xie et al., 2017; Beutel et al., 2017; Raval et al., 2017).

Bertran et al. (2019) minimize the mutual information between the utility variable and the input image data conditioned on the learned representation. Roy & Boddeti (2019) maximize the entropy of the discriminator output rather than minimizing the log likelihood, which is beneficial for stronger privacy. Osia et al. (2020) approached the problem using an information-bottleneck. Wu et al. (2018), Ren et al. (2018), and Wang et al. (2019) learn transformations of video that respect a privacy budget while maintaining performance on a downstream task. Tran et al. (2018) proposed an approach for pose-invariant face recognition. Similar to our work, their approach used adversarial learning to disentangle specific attributes in the data. Oh et al. (2017) trained a model to add a small amount of noise to the input to hide the identity of a person. Xiao et al. (2020) learn a representation from which it is hard to reconstruct the original input, but from which it is possible to predict a predefined task. The method provides control over which attributes that is preserved, but no control over which attributes that are being censored. That is, it puts more emphasis on preserving utility than privacy, which is not always desired.

All of these, with the exception of Edwards & Storkey (2016) (see below), depend on knowing the downstream task labels. *Our work has no such dependency: the data produced by our method is designed to be usable regardless of downstream task.*

In Edwards & Storkey (2016), a limited experiment is included which does not depend on the downstream task. In this experiment, they remove sensitive text which was overlaid on images, a task which is much simpler than the real-world problem considered in the current work. The overlaid text is independent of the underlying image, and therefore the solution does not require a trade-off between utility and privacy which is the case in most real settings. Furthermore, we also replace the sensitive information with synthetic information which we show further strengthens the privacy.

Like in the current work, Huang et al. (2017, 2018) use adversarial learning to minimize the mutual information between the private attribute and the censored image under a distortion constraint. Our solution extends and improves upon these ideas with a modular design consisting of a filter that is trained to obfuscate the data, and a generator that further enhances the privacy by adding new independently sampled synthetic information for the sensitive attributes.

# 3 PRIVACY-PRESERVING ADVERSARIAL REPRESENTATION LEARNING

In the current work, we propose a novel solution for utility-preserving privacy-enhancing transformations of data: we use privacy-preserving representation learning to obfuscate information in the input data, and output results that retain the information and structure of the input.

## 3.1 PROBLEM SETTING

Generative adversarial privacy (GAP) (Huang et al., 2018) was proposed as a method to provide privacy in images under a distortion constraint, and will be used as the baseline in the current work. In GAP, one assumes a joint distribution $P(X, S)$ of public data points $X$ and sensitive private attributes $S$ where $S$ is typically correlated with $X$. The authors define a privacy mechanism $X' = f(X, Z_1)$ where $Z_1$ is the source of noise or randomness in $f$. Let $h_f(X')$ be an adversary's prediction of the sensitive attribute $S$ from the privatized data $X'$ according to a decision rule $h_f$. The performance of the adversary is thus measured by a loss function $\ell_f(h_f(f(x, z_1)), s)$ and the expected loss of the adversary with respect to $X, S$ and $Z_1$ is

$$L_f(h_f, f) = \mathbb{E}_{\substack{x,s \sim p(x,s) \\ z_1 \sim p(z_1)}}[\ell_f(h_f(f(x, z_1)), s)], \tag{1}$$

where $p(z_1)$ is the source of noise.

The privacy mechanism $f$ will be trained to be privacy-preserving and utility-preserving. That is, it should be hard for an analyst to infer $S$ from $X'$, but $X'$ should be minimally distorted with respect to $X$. Huang et al. (2018) formulate this as a constrained minimax problem

$$\min_f \max_{h_f} -L_f(f, h_f) \quad \text{s.t.} \quad \mathbb{E}_{\substack{x,s \sim p(x,s) \\ z_1 \sim p(z_1)}}[d(f(x, z_1), x)] \le \epsilon_1, \tag{2}$$

where the constant $\epsilon_1 \ge 0$ defines the allowed distortion for the privatizer and $d(\cdot, \cdot)$ is some distortion measure.

In the current work, $f$ will be referred to as the *filter* since the purpose of it is to filter the sensitive information from $x$. A potential limitation with this formulation is that it only obfuscates the sensitive information in $x$ which may make it obvious to the adversary that $x'$ is a censored version of $x$. Instead, we propose to replace the sensitive information with a new independent value $s'$.

## 3.2 OUR CONTRIBUTION

We extend the filter with a *generator* module $g$, defined as $X'' = g(f(X, Z_1), S', Z_2)$ where $S'$ denotes the random variable of the new synthetic value for the sensitive attribute. $Z_1$ and $Z_2$ denote the sources of randomness in $f$ and $g$ respectively. The discriminator $h_g$ is trained to predict $s$ when the input is a real image, and to predict the "$fake$" output when the input comes from $g$ as in the learning setup in Salimans et al. (2016). The objective of the generator $g(x', s', z_2)$ is to generate a new synthetic (independent) sensitive attribute $s'$ in $x'$, that will fool the discriminator $h_g$. We define the loss of the discriminator $h_g$ as

$$L_g(h_g, g) = \mathbb{E}_{\substack{x,s \sim p(x,s) \\ s' \sim p(s') \\ z_1, z_2 \sim p(z_1, z_2)}}[\ell_g(h_g(g(f(x, z_1), s', z_2)), fake)] \quad + \mathbb{E}_{x,s \sim p(x,s)}[\ell_g(h_g(x), s)], \tag{3}$$

where $p(z_1, z_2)$ is the source of noise, $p(s')$ is the assumed distribution of the synthetic sensitive attributes $s'$, $fake$ is the fake class, and $\ell_g$ is the loss function. We formulate this as a constrained minimax problem

$$\min_g \max_{h_g} -L_g(g, h_g) \quad \text{s.t.} \quad \mathbb{E}_{\substack{x,s \sim p(x,s) \\ s' \sim p(s') \\ z_1, z_2 \sim p(z_1, z_2)}}[d(g(f(x, z_1), s', z_2), x)] \le \epsilon_2, \tag{4}$$

where the constant $\epsilon_2 \ge 0$ defines the allowed distortion for the generator.

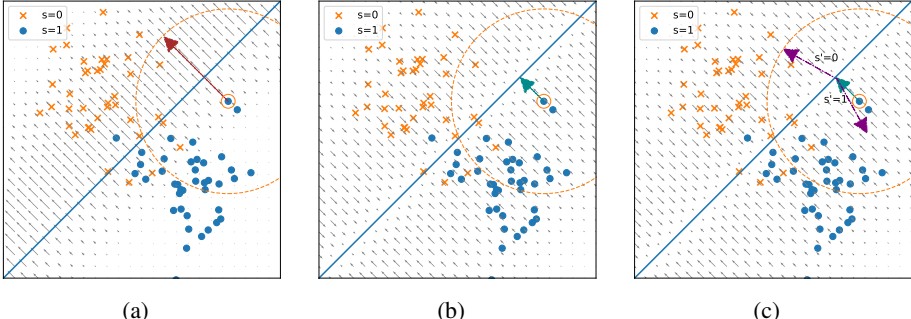

(a)  (b)  (c)

Figure 1: Visualization of privatization mechanisms. A logistic regression model is fitted to the optimal decision boundary (blue). Minimizing the log likelihood (a) leads to a model that transforms points to the opposite category. Maximizing the entropy of the adversary (b) leads to transformations that make the adversary uncertain (closer to the decision boundary). Our two-step approach (c) achieves a stronger privacy by first maximizing the entropy as in (b), and then transforming the image to a random, new value for the sensitive attribute $s$ (purple arrows). Small grey arrows show the negative gradient of the loss function with regards to the input variable, and orange circles illustrates the distortion budget.

In Figure 1 we show the difference between (a) minimizing log-likelihood of the adversary, (b) maximizing entropy of the adversary, and (c) maximizing the entropy of the adversary and also synthetically replace the sensitive attribute with a random sample.

### 3.3 IMPLEMENTATION

Let $\mathcal{D} = \{(x_i, s_i)\}_{i=1}^n$ be a dataset of samples $(x_i, s_i)$ which are assumed to be identically and independently distributed according to some unknown joint distribution $P(X, S)$. We assume that the sensitive attribute is binary and takes values in $\{0, 1\}$. However, the proposed approach can easily be extended to categorical sensitive attributes.$g(X', S', Z_2; \theta_g)$ using convolutional neural networks of the UNet (Ronneberger et al., 2015) architecture parameterized by $\theta_f$ and $\theta_g$, respectively. (See Appendix A.1 for details).

The discriminators $h_f(X'; \phi_f)$ and $h_g(X''; \phi_g)$ are modeled using ResNet-18 (He et al., 2016) and a modified version which we refer to as ResNet-10[1], respectively. The last fully connected layer has been replaced with a two and three class output layer for each model, respectively.

As suggested by Roy & Boddeti (2019), we choose the filter discriminator loss $\ell_f$ to be the the negative entropy. Intuitively, this leads to $f$ learning to make the adversary $h_f$ confused about the sensitive attribute rather than to make it certain of the wrong value. For completeness, we also include experiments where $\ell_f$ is the categorical cross-entropy, as are $\ell_{h_f}$ and $\ell_g$. The distortion measure $d$ is defined as the $L2$-norm, and $p(s')$ is assumed to be the uniform distribution $\mathcal{U}\{0, 1\}$. The hyperparameters consist of the learning rate $lr$, the quadratic penalty term coefficient $\lambda$, the distortion constraint $\epsilon$, and the $(\beta_1, \beta_2)$ parameters to Adam (Kingma & Ba, 2014). Details of the training setup can be found in Appendix A.3, and the full code is published on GitHub[2].

## 4 EXPERIMENTS

In this section we describe our experiments, the datasets used, and the evaluation metrics.

**Synthetic data.** We introduce and apply the method on a synthetic dataset to illustrate the difference between optimizing the filter for log-likelihoood and entropy, and why adding the generator is important. The synthetic data consists $\mathcal{D}_{train} = \{(x_i, s_i)\}_{i=1}^n$, $x \in \mathbb{R}^2$ and $s \in \{0, 1\}$, drawn from two different normal distributions $\mathcal{N}(\mu_1, \sigma_1)$ and $\mathcal{N}(\mu_2, \sigma_2)$, where $\mu_1 = [-1, 1]$, and $\sigma_1 =$

---

[1]ResNet-10 has the same setup as ResNet-18, but each of the "conv2_x", "conv3_x", "conv4_x", and "conv5_x" layers consists of only one block instead of two.

[2]https://github.com/anonymous/anonymous-repo-name

$[[0.7, 0][0, 0.7]]$, and $\mu_2 = [1, -1]$, and $\sigma_2 = [[0.7, 0][0, 0.7]]$. Similarly, we construct a test set $\mathcal{D}_{test} = \{(x_i, s_i)\}_{i=1}^m$, and we let $n = 400,000$ and $m = 2,560$. The points are classified according to which normal distribution they belong to, and we consider this the secret attribute, i.e, $s = 0$ if $x \sim \mathcal{N}(\mu_1, \sigma_1)$ and, $s = 1$ if $x \sim \mathcal{N}(\mu_2, \sigma_2)$. We sample $s' \sim \mathcal{U}(\{0, 1\})$. We run the experiment with distortion constraints $\epsilon \in \{0.1, 0.5, 1.0, 1.5, 2.0\}$ and run each experiment five times.

**CelebA.** The CelebA dataset[3] (Liu et al., 2015) consists of 202,599 face images of size 218x178 pixels and 40 binary attribute annotations per image, such as age (old or young), gender, if the image is blurry, if the person is bald, etc. The dataset has an official split into a training set of 162,770 images, a validation set of 19,867 images and a test set of 19,962 images. We resize all images to 64x64 for the quantitative experiments, and to 128x128 pixels for the qualitative experiments, and normalize all pixel values to the region $[0, 1]$. We use a higher resolution for the qualitative results to make subtle visual changes more apparent.

**Filtering and replacement of sensitive data.** Let $\mathcal{D}_{train} = \{(x_i, s_i\}_{i=1}^n$ be a set of training data where $x_i$ denotes facial image $i$ and $s_i \in \{0, 1\}$ denotes the sensitive attribute. Further let $\mathcal{D}_{test} = \{(x_i, s_i)\}_{i=1}^m$ be the held out test data. For the purpose of evaluation, we assume access to a number of utility attributes $u^{(j)} \in \{0, 1\}$ for each $x$ and $j \in U$. The following attributes provided with the dataset will be used in the CelebA experiments: *gender*, *lipstick*, *age*, *high cheekbones*, *mouth slightly open*, *heavy makeup*, *smiling*. In each experiment, one of these will be selected as the sensitive attribute. The rest will be considered the utility attributes, not used for training, but allow for a utility score in the evaluation. This score shows how well non-sensitive attributes are preserved in the transformation. We compute an average over fixed classifiers, trained to predict each respective utility attribute on the original training data, when evaluated on the censored data.

**Hyperparameters.** We train the models using $\mathcal{D}_{train}$ with $lr = 0.0005$, $\lambda = 10^5$, $\epsilon \in \{0.03, 0.02, 0.01, 0.005, 0.001\}$, and $(\beta_1, \beta_2) = (0.9, 0.999)$. Let the training data censored by the filter be $\mathcal{D}'_{train} = \{(x'_i, s_i)\}_{i=1}^n$, where $x'_i = f(x_i, z_i^{(1)}; \theta_f)$, and by both the filter and the generator be $\mathcal{D}''_{train} = \{(x''_i, s_i)\}$ where $x''_i = g(x'_i, s'_i, z_i^{(2)}; \theta_g)$, $z_i^{(1)}, z_i^{(2)} \sim \mathcal{N}(\mathbf{0}, \mathbf{1})$, and $s'_i \sim \mathcal{U}\{0, 1\}$. We do the same transformations to the test data and denote them $\mathcal{D}'_{test}$ and $\mathcal{D}''_{test}$ respectively.

**Computational requirements.** Each experiment was performed on a Tesla V100 SXM2 32 GB in a DGX-1 machine. The training was restricted to 100 epochs which takes about 13 hours. We run each experiment with negative entropy loss three times, and present the average over these runs, and we run each experiment with log-likelihood loss five times and present the average over these.

**Evaluation.** To evaluate the privacy loss for our method we train an adversarial classifier to predict the ground truth sensitive attribute given a training set of images censored by each privatization method, and then we evaluate each adversary on a test set of censored images. If an adversary can predict the ground truth sensitive attribute of censored images, that it has not seen before, with high accuracy, then the privacy loss is high. Let $adv(s|h(x))$ denote an adversary trained on the censored training set to predict the ground truth attribute $s$ given censored image $h(x)$ then the privacy loss is defined as

$$Privacy\ loss = \frac{1}{|\mathcal{D}_{test}|} \sum_{(x,s) \in \mathcal{D}_{test}} \mathbb{1}[adv(s|h(x)) = s]. \tag{5}$$

To evaluate the utility score on the CelebA data, we use fixed classifiers that have been pre-trained on the original data to predict a set of utility attributes that are considered non-sensitive. If these utility attributes are predictable with high accuracy after privatization of the image, then we consider the method to have a high utility score. Let $fix(u|x'')$ denote a fixed classifier trained on the original training set to predict the ground truth attribute $u$, then the utility score is defined as

$$(CelebA)\ \text{Utility score} = \frac{1}{|U|} \sum_{j \in U} \frac{1}{|\mathcal{D}_{test}|} \sum_{(x,u^{(j)}) \in \mathcal{D}_{test}} \mathbb{1}[fix(u^{(j)}|h(x)) = u^{(j)}]. \tag{6}$$

---

[3]http://mmlab.ie.cuhk.edu.hk/projects/CelebA.html

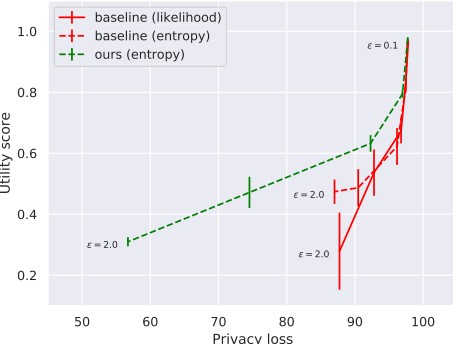

Figure 2: The trade-off curve between privacy loss and utility score for the baseline (likelihood), the improved baseline (entropy) and our method on the synthetic dataset. Upper left corner is better.

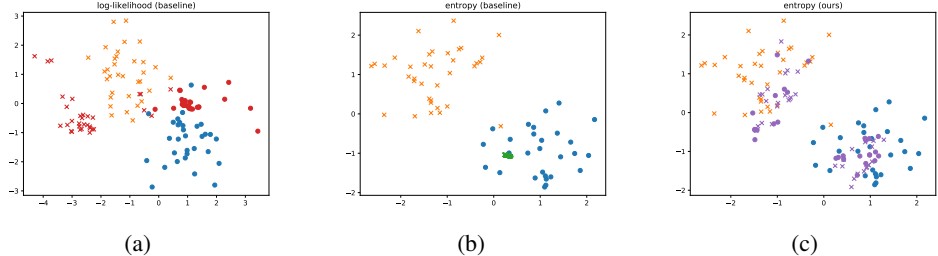

|     |     |     |
|:---:|:---:|:---:|
| (a) | (b) | (c) |

Figure 3: Results on the synthetic dataset. The censored representations by (a) baseline with filter trained to minimize log-likelihood of adversary, (b) the baseline filter trained to maximize entropy, and (c) filter trained to maximize entropy and then generating a new synthetic attribute.

To evaluate the utility score on the synthetic data, we compute

$$\textit{(Synthetic)} \text{ Utility score} = 1 - \frac{1}{\epsilon_{max}|\mathcal{D}_{test}|} \sum_{(x,s)\in\mathcal{D}_{test}} |x - h(x)|, \quad (7)$$

where we normalize with $\epsilon_{max} = 2$ to get distances into the range $[0, 1]$ and convert it to a similarity by subtracting the average over these distances from one for consistency in the trade-off curves. Where we let $h(x) = f(x, z_1; \theta_f)$ to evaluate the baseline, and $h(x) = g((f(x, z_1; \theta_f), s', z_2; \theta_g)$ to evaluate our method.

We then plot the utility score against the privacy loss for different distortion budgets and get a trade-off curve. Similar evaluation method are used in Roy & Boddeti (2019); Bertran et al. (2019). In particular, Roy & Boddeti (2019) plot accuracy of an adversary predicting the sensitive attribute against the accuracy of a classifier predicting the target attribute.

## 5 RESULTS

In this section we present quantitative and qualitative results on the facial image experiments.

**Synthetic data.** Figure 2 shows the trade-off between privacy loss and utility score for the experiment on the synthetic dataset. Our method consistently outperforms the baseline for all given distortion budgets. Figure 3 shows the original data (orange for s=0, and blue for s=1) together with the censored data for the three different transforms (red for baseline with likelihood, green for baseline with entropy, and purple for the proposed method). When minimizing log-likelihood we see clear clusters on opposite side of the decision boundary, when maximizing entropy we see that all points approach the decision boundary, and when adding the generator we see that the points

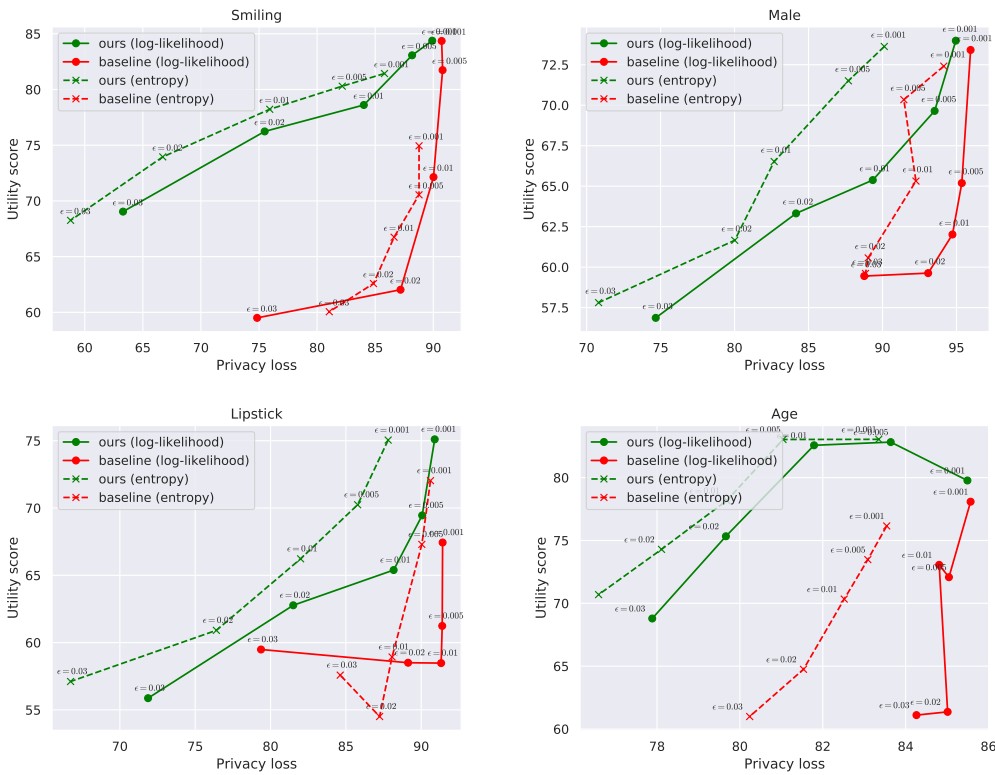

Figure 4: Privacy vs. utility trade-off curve where the sensitive attribute is *smiling* (top left), *gender* (top right), *lipstick* (bottom left), *age* (bottom right). Our approach with negative entropy loss consistently outperforms all other approaches on the attributes explored, and our method with log-likelihood loss outperforms the baseline with log-likelihood loss on all explored attributes.

are mapped, at random, into the cluster of orange points or blue points depending on the sampled synthetic sensitive attribute.

**CelebA.** Figure 4 shows the trade-off between privacy loss and utility score when evaluated on censored images. Our method consistently has a higher utility at any given level of privacy compared to the baseline. Remember: these are strong adversaries required to run tagged training data through the privacy mechanism to be able to train. Additional results can be found in Appendix A.2.

To further show that explicitly optimizing for privacy in the privatization mechanism is necessary we have conducted a similar experiment using StarGAN Choi et al. (2018) to randomly change the sensitive attribute in the image (results in suppl.). We then evaluate the censored images, which look very convincing to a human, using adversarial classifiers. The adversaries can successfully detect the sensitive attributes with an accuracy of roughly 90%. For the weight $\lambda_{rec}$ of the cycle consistency loss we explored values $\{0, 5, 10, 50\}$. We obtain similar scores when we exclude the filter part of our method and use only the generator part to censor the images (see Appendix A.2).

In Table 1 we present the results of evaluating the accuracy of $fix(s|\cdot)$ on the dataset $\{x_i'', s_i'\}_{i=1}^m$ where $x_i'' = g(x_i', s_i' z_i^{(2)})$ is the image censored with our method and $s_i'$ is the new synthetic attribute uniformly sampled from $\{0, 1\}$. That is, we measure how often the classifier predict the new synthetic attribute $s_i'$ when applied to $x_i''$. We can see that with $\epsilon = 0.001$ the method is on average able to fool the classifier $82.4\%$ of the time for the *smiling* attribute, and this increases with larger distortion budget $\epsilon$ to a success rate of $91.2\%$ on average with $\epsilon = 0.05$. The results are similar when the images have been censored with respect to the attributes *gender*, *lipstick*, and *age*, but these require a larger distortion budget.

Table 1: The success rate of our method to fool a fixed classifier for the sensitive attributes *smiling*, *gender*, *lipstick*, and *age*. This was measured as the accuracy of $fix(s|\cdot)$ for each censored attribute in the censored data $\{(x_i'', s_i')\}_{i=1}^m$. Higher is better. Average and standard deviation of five runs.

| Dist. | | Synthetic | | |
|---|---|---|---|---|
| $\epsilon$ | Smiling | Gender | Lipstick | Young |
| 0.001 | $82.4 \pm 2.1$ | $72.9 \pm 1.5$ | $62.2 \pm 3.1$ | $59.2 \pm 1.9$ |
| 0.005 | $86.4 \pm 3.7$ | $79.4 \pm 0.6$ | $71.7 \pm 2.4$ | $62.1 \pm 2.5$ |
| 0.01 | $87.4 \pm 2.1$ | $85.6 \pm 1.4$ | $77.6 \pm 2.4$ | $59.6 \pm 1.6$ |
| 0.05 | $91.2 \pm 2.9$ | $90.3 \pm 4.3$ | $90.6 \pm 4.2$ | $67.7 \pm 2.1$ |

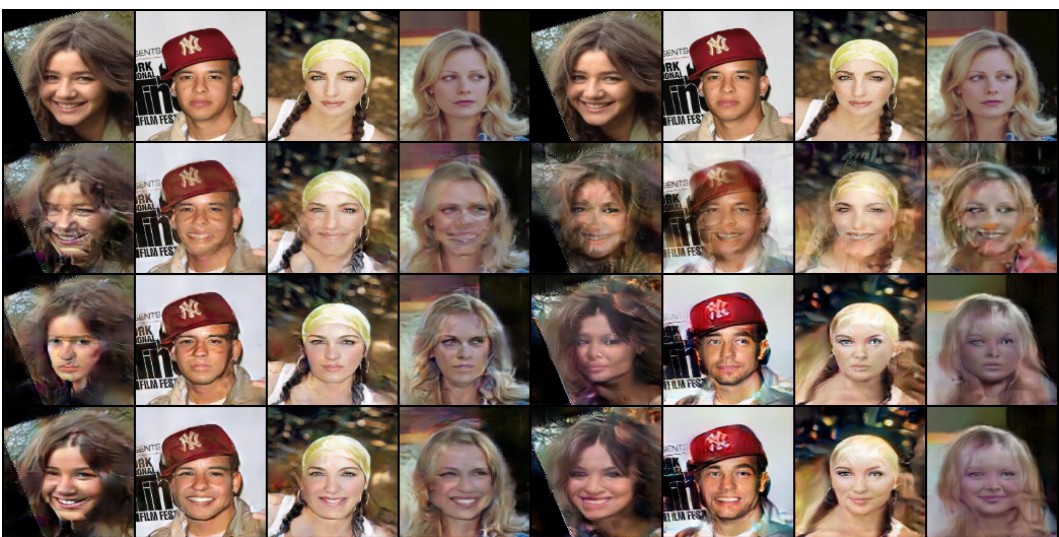

Figure 5: Qualitative results for the sensitive attribute smile. In the first four columns: $\epsilon = 0.001$, and in the last four columns: $\epsilon = 0.01$. From top to bottom row: input image ($x$), censored image ($x'$), censored image with synthetic non-smile ($x''$, $s' = 0$), censored image with synthetic smile ($x''$, $s' = 1$). The model is able to generate a synthetic smiling attribute while maintaining much of the structure in the image. These images were generated from a model trained using 128x128 pixels.

Figure 5 shows, from the top row to the bottom row, the input image $x$, the censored image $x'$, the censored image $x''$ with the synthetic attribute $s' = 0$ (non-smiling), and the censored image $x''$ with the synthetic attribute $s' = 1$ (smiling). A value of $\epsilon = 0.001$ is used in the first four columns, and $\epsilon = 0.01$ in the last four columns. The images censored by our method look sharper and it is less obvious that they are censored. We can see that the method convincingly generates non-smiling faces and smiling faces while most of the other parts of the image is intact. These images are sampled from models trained on images of 128x128 pixels resolution. See Figure 7 in Appendix A.2 for corresponding samples on the same input images, but using *gender* as the sensitive attribute.

## 6 DISCUSSION

In Figure 1, we show how a privatization mechanism is affected by an adversarial setup and its training criterion. Minimizing the log likelihood of the adversary can be interpreted as attempting to make the adversary certain of the wrong value of the sensitive attribute and may lead to transforming data with a certain sensitive value to be transformed to similar outputs. Following the gradient of the entropy loss instead leads to the adversary becoming less certain and thus leads to more privacy. Using this approach to remove sensitive information, and then adding random information makes an even stronger privacy, as we take another step adding randomness to the output. This intuition

is confirmed in the experimental results on the synthetic data, as demonstrated by Figure 3. Our approach successfully transforms the datapoints into a distribution which is nearly indistinguishable from the input distribution, and where datapoints with the different sensitive values are completely mixed.

Our method consistently outperforms the baseline, ensuring a higher level of privacy while maintaining more of the useful information (see Figure 4). For all sensitive attributes that we consider and at nearly all given distortion budgets ($\epsilon \in \{0.02, 0.01, 0.005, 0.001\}$) we observe both a higher privacy and a higher utility for our method. For some of the attributes (*gender*, *lipstick*), at extreme distortions ($\epsilon = 0.03$) our method has a utility score which is not better than the baseline, but still provide considerably better privacy. Furthermore, we can observe that using the entropy loss function for the filter benefits both the baseline and our method.

This shows that our method makes it more difficult for the adversary to see through the privatization step at each given distortion budget. To show the effect of the filter we have conducted experiments where we only use the generator to privatize the images (see Appendix A.2), which does not seem to provide any privacy.

A generator without the filter may learn to either pass the original image without modification (when $s' = s$) or to transform the image into a random other value ($s' \neq s$). If the transformed image is indistinguishable from a real image this is not a problem, but otherwise we can easily reverse the privatization by detecting if the image is real or not. The filter step mitigates this by *always* removing the sensitive data in the image, forcing the generator to synthesize new data. Since the censored image is now guaranteed to be synthetic, we are no longer susceptible the simple real/fake attack.

Similarily, we conduct experiments where we use StarGAN to censor the images, but observe no privacy when using this standard attribute manipulation method. A reason could be that the cyclic consistency loss term actually encourage image transformations that are easily invertible given the original attribute, which is not desirable in a privacy setting. This motivates our approach which is explicit about the privacy objective.

In Table 1, we see that the fixed smile classifier is fooled by our privatization mechanism in 82.4% to 91.2% of the data points in the test set (depending on the distortion $\epsilon$). These results indicate that it may be harder for an adversarially trained classifier to predict the sensitive attribute when it has been replaced with something else, as compared to simply removed. We assume that this is due to the added variability in the data. Or intuitively: it is easier to "blend in" with other images that have similar demonstrations of smiles.

The fact that many important attributes in facial images correlate leads to the reflection that disentangling the underlying factors of variation is not entirely possible. For example, in CelebA, lipstick is highly correlated with female gender. This means that if we want to hide all information about the presence of lipstick we also need to hide the gender (and other correlating attributes). This problem is further analysed in Table 3 in Appendix A.2.

A strength of our method is that it is *domain-preserving*, this allows a utility provider to use the censored image in existing algorithms without modifications. Since the method also preserves utility this may also allow stacking the privatization mechanism to censor multiple attributes in an image.

## 7   CONCLUSIONS

In this work we have presented a strong privacy-preserving transformation mechanism for image data which is learned using an adversarial setup. While previous work on adversarial representation learning has focused on removing information from a representation, our approach extends on this and can also generate new information in its place that looks realistic and gives further privacy properties compared to the baseline. We evaluate our method using adversarially trained classifiers, and our results show that not only do we provide stronger privacy with regards to sensitive attributes, but we also preserve non-sensitive attributes of the image at a higher rate. The results show that the synthetically added attribute gives stronger privacy properties and helps fooling the adversary in the challenging setting where the adversary is allowed to be trained using the output of the privacy mechanism.

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

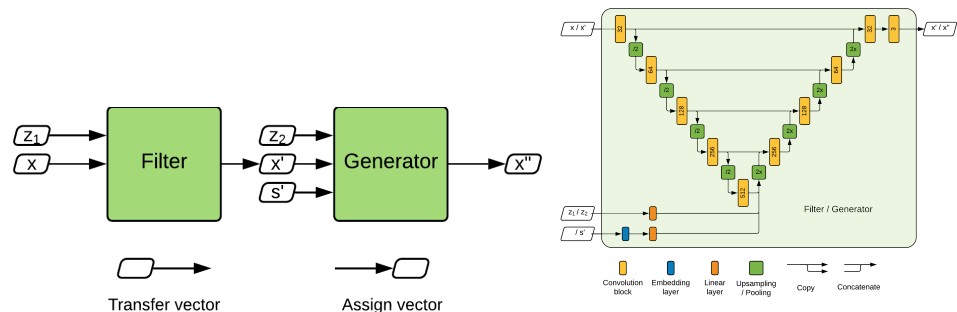

Figure 6: Overview of the training setup (left) and the network architecture used in the filter and the generator (right).

## A    APPENDIX

### A.1    ADDITIONAL DETAILS

An overview of the setup can be seen in Figure 6. We use the UNet Ronneberger et al. (2015), illustrated on the right in Figure 6, architecture for both the filter and the generator. The orange blocks are convolution blocks each of which, except for the last block, consist of a convolution layer, a batch normalization layer and a rectified linear activation unit, repeated twice in that order. The number of output channels of the convolution layers in each block has been noted in the figure. The last convolution block with a 3 channel output (the RGB image) consists of only a single convolutional layer followed by a sigmoid activation. The green blocks denote either a max pooling layer with a kernel size of two and a stride of two if marked with "/2" or a nearest neighbor upsampling by a factor of two if marked with "2x". The blue block denotes an embedding layer, which takes as input the categorical value of the sensitive attribute and outputs a dense embedding of 128 dimensions. It is then followed by a linear projection and a reshaping to match the spatial dimensions of the output of the convolution block to which it is concatenated, but with a single channel. The same type of linear projection is applied on the 1024 dimensional noise vector input, but this projection and reshaping matches both the spatial and channel dimensions of the output of the convolutional block to which it is concatenated. Concatenation is in both cases done along the channel dimension.

### A.2    ADDITIONAL RESULTS

The results in Table 2 show that using the filter together with the generator ($g \circ f$) give a stronger privatization guarantee than using only the generator (without the filter) or only the baseline (the filter without the generator).

Table 2: The results of evaluating the adversarially trained classifiers $adv(s|\cdot)$ on the held out test data censored with the baseline, only the generator, and our method for varying distortion $\epsilon$ for the *smiling* attribute. Closer to 50 means more privacy. Only using the generator does not privide any privacy, but using it together with the filter provides the best privacy. Lower is better.

| | **Adversarial smiling** | | |
| Dist. $\epsilon$ | baseline ($f$) | generator ($g$) | ours ($g \circ f$) |
| --- | --- | --- | --- |
| 0.001 | 89.3 | 90.7 | 88.7 |
| 0.005 | 89.9 | 91.2 | 85.9 |
| 0.01 | 89.8 | 91.5 | 83.2 |
| 0.05 | 69.2 | 78.7 | 54.1 |

In Figure 7 we show additional qualitative results when the attribtue *gender* is considered sensitive.

Table 3 shows the correlations between classifier predictions on a pair of attributes when one attribute has been synthetically replaced.

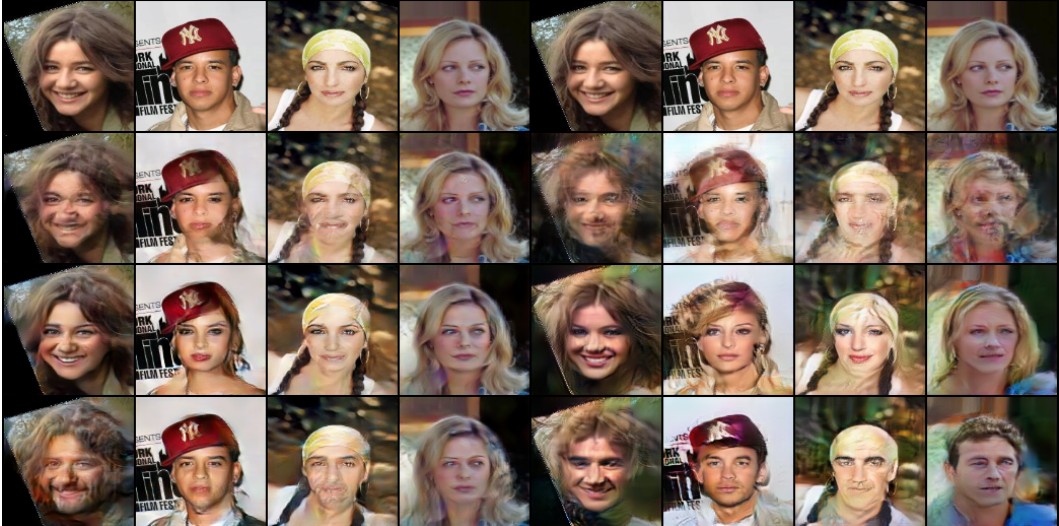

Figure 7: Qualitative results for the sensitive attribute gender. In the first four columns: $\epsilon = 0.001$, and in the last four columns: $\epsilon = 0.01$. From top to bottom row: input image ($x$), censored image ($x'$), censored image with synthetic female gender ($x''$, $s' = 0$), censored image with synthetic male gender ($x''$, $s' = 1$). The model is able to generate a synthetic gender while maintaining much of the structure in the image. These images were generated from a model trained using 128x128 pixels.

Table 3: The value of each cell denotes the Pearson's correlation coefficient between predictions from a classifier trained to predict the row attribute and a classifier trained to predict the column attribute, given that the column attribute has been censored.

|  | Smiling | Gender | Lipstick | Young |
|---|---|---|---|---|
| Smiling | 1.00 | -0.04 | 0.08 | -0.06 |
| Gender | -0.07 | 1.00 | -0.44 | -0.21 |
| Lipstick | 0.14 | -0.30 | 1.00 | 0.26 |
| Young | 0.05 | -0.11 | 0.23 | 1.00 |
| High Cheekbones | 0.14 | -0.07 | 0.15 | -0.01 |
| Mouth Open | 0.04 | 0.00 | 0.03 | -0.02 |
| Heavy Makeup | 0.12 | -0.24 | 0.47 | 0.22 |

For example, in the CelebA dataset lipstick is highly correlated with female gender. This means that if we want to hide all information about whether or not the person is wearing lipstick we also need to hide its gender (and other correlating attributes). This problem can be seen in Table 3 where changing whether or not a person is wearing lipstick correlates with changes of gender.

The question is: if we censor an attribute in an image, how does that correlate with changes of other attributes in the image? In the lipstick column of Table 3 we have censored the attribute lipstick. We then make predictions on whether or not the person in the censored image is wearing lipstick, and compute the correlation between these predictions and predictions for the attributes for each row. We can see that changes in lipstick correlate negatively with changes in gender and positively with makeup. This highlights the problem of disentangling these underlying factors of variation.

Table 4 shows the Pearson correlations between the smiling attribute and 37 other attributes in the CelebA dataset.

Table 4: Pearson correlation coefficient between the smiling attribute and 37 other attributes in the CelebA training dataset, ordered from high to low absolute correlation.

| Attribute | Correlation |
|---|---|
| High cheekbones | 0.68 |
| Mouth slightly open | 0.53 |
| Rosy cheeks | 0.22 |
| Oval face | 0.21 |
| Wearing lipstick | 0.18 |
| Heavy makeup | 0.18 |
| Wearing earrings | 0.17 |
| Attractive | 0.15 |
| Gender | -0.14 |
| Bags under eyes | 0.11 |
| No beard | 0.11 |
| Big nose | 0.10 |
| Double chin | 0.10 |
| Arched eyebrows | 0.09 |
| Wearing necklace | 0.09 |
| Blond hair | 0.09 |
| Narrow eyes | 0.08 |
| Sideburns | -0.08 |
| Wavy hair | 0.08 |
| Mustache | -0.07 |
| Pale skin | -0.07 |
| Five-o'clock shadow | -0.07 |
| Goatee | -0.07 |
| Blurry | -0.06 |
| Wearing hat | -0.06 |
| Bangs | 0.05 |
| Chubby | 0.04 |
| Eyeglasses | -0.04 |
| Pointy nose | 0.04 |
| Brown hair | 0.02 |
| Receding hairline | 0.02 |
| Bald | 0.01 |
| Big lips | 0.01 |
| Gray hair | 0.01 |
| Straight hair | 0.01 |
| Black hair | -0.00 |
| Bushy eyebrows | -0.00 |
| Wearing necktie | -0.00 |

## A.3 TRAINING SETUP

Algorithm 1 demonstrates the training setup for our method. Note that when $\ell_f$ is negative entropy it does not depend on the $1 - s_i$ argument, but is simply computed from the output of the discriminator $h_f$. However, when $\ell_f$ is cross entropy we optimize the filter such that the discriminator $h_f$ is fooled into predicting the complement class $1 - s$. This means that we need to assume $s$ to be a binary attribute when using this loss. When using the negative entropy we do not need to make this assumption, which is a strength of the proposed method.

**Algorithm 1**

**input:** $\mathcal{D}, lr, \lambda, \epsilon, \beta_1, \beta_2$
$\epsilon_1, \epsilon_2 \leftarrow \epsilon$
**repeat**

    Draw m samples uniformly at random from the dataset
    $(x_1, s_1), \ldots, (x_m, s_m) \sim \mathcal{D}$
    Draw m samples from the noise distribution
    $(z_1^{(1)}, z_1^{(2)}), \ldots, (z_m^{(1)}, z_m^{(2)}) \sim p(z^{(1)}, z^{(2)})$
    Draw m samples from the synthetic distribution
    $s_1', \ldots, s_m' \sim p(s')$
    Compute censored and synthetic data
    $x_1', \ldots, x_m' = f_{\theta_f}(x_1, z_1^{(1)}), \ldots, f_{\theta_f}(x_m, z_m^{(1)})$
    $x_1'', \ldots, x_m'' = g_{\theta_g}(x_1', s_1', z_1^{(2)}), \ldots, g_{\theta_g}(x_m', s_m', z_m^{(2)})$
    Compute filter and generator losses

$$\Theta_f(\theta_f) = \frac{1}{m} \sum_{i=1}^{m} \ell_f(h_f(x_i'; \phi_f), 1 - s_i)$$
$$+ \lambda \max(\frac{1}{m} \sum_{i=1}^{m} d(x_i', x_i) - \epsilon, 0)^2$$

$$\Theta_g(\theta_g) = \frac{1}{m} \sum_{i=1}^{m} \ell_g(h_g(x_i''; \phi_g), s_i')$$
$$+ \lambda \max(\frac{1}{m} \sum_{i=1}^{m} d(x_i'', x_i) - \epsilon, 0)^2$$

    Update filter and generator parameters
    $\theta_f \leftarrow \text{Adam}(\Theta_f(\theta_f); lr, \beta_1, \beta_2)$
    $\theta_g \leftarrow \text{Adam}(\Theta_g(\theta_g); lr, \beta_1, \beta_2)$
    Compute discriminator losses
    $\Phi_f(\phi_f) = \frac{1}{m} \sum_{i=1}^{m} \ell_{h_f}(h_f(x_i'; \phi_f), s_i)$

$$\Phi_g(\phi_g) = \frac{1}{m} \sum_{i=1}^{m} \ell_g(h_g(x_i''; \phi_g), fake)$$
$$+ \frac{1}{m} \sum_{i=1}^{m} \ell_g(h_g(x_i; \phi_g), s_i)$$

    Update discriminator parameters
    $\phi_f \leftarrow \text{Adam}(\Phi_f(\phi_f); lr, \beta_1, \beta_2)$
    $\phi_g \leftarrow \text{Adam}(\Phi_g(\phi_g); lr, \beta_1, \beta_2)$
**until** stopping criterion
**return** $\theta_f, \theta_g, \phi_f, \phi_g$

