# OpenReview forum: "Adversarial representation learning for synthetic replacement of private attributes"
_ICLR.cc/2021/Conference — Reject_

### Official Review · AnonReviewer3 · 2020-10-28
**interesting study, unconvincing experiments**

**Rating:** 5
**Confidence:** 3

**Review:**

Summary:
This work proposes a privacy-preserving transformation mechanism based on adversarial representation learning. The proposed work is an extension of generative adversarial privacy (GAP). It replaces the filter with a generator that replaces the sensitive information with synthetic data instead of censoring it.

Reason for score:
The contribution of the paper is incremental, and the experimental results are limited to one dataset and one baseline. More experiments need to be done to evaluate the consistency of the results and to study the generalizability of this work across various datasets.

More detailed comments:
-         In related work, please add more recent works in adversarial representation learning and explain the contribution of this work compared to the existing ones. One example includes:
Adversarial Learning of Privacy-Preserving and Task-Oriented Representations by Xiao et al.
-         It would be better if the authors theoretically prove the privacy-preserving of the proposed method.
-         This work is only evaluated on one dataset, also it is only compared to one baseline. The authors are expected to compare the proposed method with several recent works (some of them already mentioned in related work) and show how this method advances the state-of-the-art in this area.
-         Please explain how this work can scale on larger data or more high-quality images? More discussion on the efficiency of the method is also necessary compared to other existing techniques.
-         It would be better if the authors showed how the model preserves utility when generating  multiple synthetic attributes especially in cases where two or more attributes are highly correlated (e.g., heavy makeup and lipstick or mouth open and smiling, etc.)

---

> ### Author Response · Authors · 2020-11-18
> **Response to AnonReviewer3**
>
> Thank you for reviewing our paper. We have updated the paper in response to your valuable feedback.
>
> In particular, we have added a thorough discussion about different learning paradigms in the adversarial learning setup, and how the learning objectives affect the learned mechanism (see new Figure 1). Furthermore, we have improved the evaluation and added a new dataset that confirms this discussion (see new Figure 2 and 3). From this discussion and evaluation, one can clearly see how the entropy loss helps learning a stronger privatization mechanism, and how adding the second step with independent synthetic attributes further strengthens the privacy. In addition to the baseline as described by Huang et.al. (2018), we also have an improved baseline which uses the entropy loss. To the best of our knowledge, most of the prior work relies on knowing the downstream task and is optimized for that. We make no such assumption on the task labels, and study the setting where utility is preserved through a constraint on the distortion budget allowed for the privatization mechanism. The exception is the pioneering work of Edwards et. al. and the work of Huang et. al.. Edwards et al, in their experiment on privatization of images without dependence on the downstream task, consider a very similar optimization setup as Huang et. al, the difference is that they treat it as a weighted sum between the distortion and the privacy, and not a constrained optimization of privacy subject to a distortion budget. We do not expect large differences in the achievable trade-off between privacy and utility between the constrained optimization in Huang et. al, and the weighted sum studied by Edwards et. al., the main difference is that the constrained optimization makes the choice of the parameter that controls the trade-off a bit more intuitive.
>
> We have added recent related work to the paper, see page 2.
>
> We have a more thorough discussion about the benefits of the proposed approach on a fundamental level. See the Discussion section on page 8 and 9. Together with the exploration of different learning objectives and their effects, and the added experimental evaluation with a new dataset as mentioned above, your comment has really helped us improve our paper.
>
> As you request, we have a discussion about the computational requirements for the method on page 5.
>
> Your comment on “multiple synthetic attributes” could be interesting. However, we have deemed it more interesting to see the performance of the setting where we censor one specific attribute as this makes the evaluation clearer, and the problem is actually also harder this way. As all attributes except the sensitive one are considered utility attributes, the problem of disentangling attributes which are strongly correlated adds to the difficulty of the problem. Heavy make-up and lipstick are certainly highly correlated, and censoring both at the same time would only make it easier for our approach. With that said, the setting of censoring more than one attribute at a time is certainly an interesting application where I'm sure our method would be applicable, either by training the adversary to detect all sensitive attributes at once, or by stacking several privacy mechanisms together.

---

### Official Review · AnonReviewer1 · 2020-10-28
**Review of "Adversarial representation learning for synthetic replacement of private attributes"**

**Rating:** 4
**Confidence:** 4

**Review:**

Summary: The paper presents a privacy-preserving transformation technique for image data. The main idea is to use adversarial representation learning to obfuscate sensitive attributes.

Strengths:
1) The paper considers an important problem of preserving privacy on images and presents a heuristic approach to address this problem.
2) The idea of using adversarial representation learning for this problem seems reasonable. The prior work on this topic has  focused on removing information from representation, and the paper extends this to also generating new information.

Concerns:
1) The paper feels incremental; the idea utilized seems like a simple extension to the Generative adversarial privacy idea of Huang et al.
2) The evaluation feels a little strange. The authors consider a set of attributes gender, lipstick, age, high cheekbones, mouth slightly open, heavy makeup, smiling (and split them as sensitive and utility). I did not follow the real motivation of using these set of attributes, why are they sensitive? A better dataset (usecase) and/or choice of attributes would have helped making a stronger case. Being an experimental paper, a stronger evaluation (say on multiple datasets) might also help.

Overall, while I feel that the problem is interesting, the paper presents just too little in terms of its technical contribution.

---

> ### Author Response · Authors · 2020-11-18
> **Response to AnonReviewer1**
>
> Thank you for your insightful comments and valued feedback.
>
> While we agree on the importance of the problem and that the adversarial setup is reasonable, we argue that using an adversarial setup is a way of not having to rely on heuristics (such as detecting the bounding box of a face in an image, assume that these are the private pixels, and changing only those pixels, as some approaches do). If there is a weakness in the privacy mechanism in the adversarial setup,  the adversary will most likely detect them, which in this learning paradigm would actually just be used to improve the privacy mechanism in the next iteration. Hence, this is an approach that could lead to really strong and robust results.
>
> We have revised the paper with the guidance of your review. You are right that Huang, et.al (2018) made crucial progress on adversarial learning for privacy. However, our paper does contain important insights, and a substantial technical contribution which we hope to have made clearer with the following additional results and analysis. We have added a thorough discussion about the effects of different learning objectives and the benefits of the proposed learning setup (see new Figure 1). This shows that the entropy objective together with the added independent synthetic attributes make a substantial contribution to the level of achieved privacy at a foundational level. We demonstrate this using a detailed discussion about the properties of the learning criteria and have also added new experiments on a new dataset to strengthen our claims (see new Figure 2 and 3). This analysis and these results also show a crucial weakness in the objective by Huang et.al.: While their approach manages to transform data so that it is not recognisable with the same means of detection, the censored data still contains identifying characteristics.
>
> The evaluation has been strengthened as we now demonstrate results on two different datasets (see above). The CelebA dataset is widely used in the literature and the considered sensitive attributes are in fact interesting in that they pose difficult privatization problems, with high correlation between the sensitive and the non-sensitive attributes: the disentanglement of these is a fundamentally difficult problem and we do show important progress with this work.

---

### Official Review · AnonReviewer4 · 2020-10-29
**Weak Reject**

**Rating:** 5
**Confidence:** 2

**Review:**

The paper introduces a framework to privatize sensitive attributes of data using adversarial representation learning. The proposed method consists of a “filter” that removes the sensitive attribute from the data representation, and a “generator” that replaces the removed sensitive attribute with a randomly sampled synthetic value. The authors argue that the second step done by the generator enhances privacy, and use experiments on real image data to verify their method and compare it with a baseline.

One important aspect of this work is that the privatization scheme is done regardless of the downstream learning task. The authors argue that all prior work (except for one) doesn’t have this property.

Some comments and questions:

-Can you provide more evidence as to why replacing the sensitive information with something else (which is what the generator does) is useful? In the third paragraph of introduction you mention this as an “assumption” but is this actually an assumption or observation?

-In your experiments I couldn’t find what the baseline is? Is it just the Huang et al. 2018 approach which is the same method without the generator? Are there any other methods you could compare your method with?

-In your experiments, why is it that in some cases when epsilon is increased, both privacy loss and utility score gets better? For e.g. see Age plot in Figure 1, epsilon=0.001 and epsilon=0.005 for the green curve. Shouldn’t we always have that increasing epsilon helps the privacy loss but hurts the utility score?

-In Eq. 5 & 6, Z_2 is probably missing inside the parentheses for g.

---

> ### Author Response · Authors · 2020-11-18
> **Response to AnonReviewer4**
>
> Thank you for an excellent and insightful review of our paper.
>
> We have updated the paper in response to your concerns.
>
> In particular, we provide new evidence for the usefulness of replacing the private information with independent random information. This is highlighted with a discussion around the learning objectives of the different setups, together with visualizations of the criteria in a low-dimensional setting (see new Figure 1). This argument is accompanied with new experiments on a new synthetically generated dataset which confirm our premise and strengthen the conclusions of the paper (see new Figures 2 and 3).
>
> The main baseline is as you state, the filter method from Huang et.al. (2018), but we have also created a new baseline which is stronger. This one is also one filter module but it is trained using the entropy criterion which improves the privacy substantially. This is also highlighted by the additions discussed in the previous paragraph.
>
> Regarding: “Shouldn’t we always have that increasing epsilon helps the privacy loss but hurts the utility score?”
>
> This is a valid question, and in most cases our data also confirms your assumption. However, interestingly, we also see some situations where the results go against this intuition. We believe that this may have to do with a regularizing effect that censoring a sensitive attribute may have for other attributes in the data. In the dataset, there are many attributes that correlate (positively or negatively) with each other. In particular, age correlates with lipstick, heavy make-up, and gender. We conjecture that making transformations that give a random variation in age may lead to the model learning to more clearly convey this correlated information from the original image. This is of course learned entirely without any supervision; the utility attributes are only used for evaluation, never for training. In the domain adaptation literature (e.g. Kim, et.al., Learning not to learn: Training Deep Neural Networks With Biased Data, CVPR 2019), there are discussions about how learning an invariance to certain correlating attributes, can lead to a model learning the desired task even better, and the adversarial setup they study to learn such invariances is in fact quite similar to our setup where we basically learn a representation (censored image) which is invariant towards the private attribute, and in some instances such an invariance may possibly lead to better performance for other tasks.

---

### Decision · Program_Chairs · 2021-01-07
**Final Decision**

**Decision:**

Reject

**Comment:**

This paper proposes a heuristic for removing privacy sensitive attributes and replacing them with sythetically generated ones. The technique is closely related to an existing work and, as pointed out in the reviews, the experimental evaluation is insufficient for properly evaluating the approach.